# Immunoprofiling Identifies Functional B and T Cell Subsets Induced by an Attenuated Whole Parasite Malaria Vaccine as Correlates of Sterile Immunity

**DOI:** 10.3390/vaccines10010124

**Published:** 2022-01-14

**Authors:** Marie Mura, Pinyi Lu, Tanmaya Atre, Jessica S. Bolton, Elizabeth H. Duncan, Sidhartha Chaudhury, Elke S. Bergmann-Leitner

**Affiliations:** 1Biologics Research and Development, Walter Reed Army Institute of Research, Silver Spring, MD 20910, USA; marie.mura@intradef.gouv.fr (M.M.); tanmaya.atre@bcchr.ca (T.A.); jessica.s.bolton.ctr@mail.mil (J.S.B.); elizabeth.h.duncangooden.civ@mail.mil (E.H.D.); 2Immunopathology, Institut de Recherche Biomédicale des Armées, 91223 CEDEX, BP73 Brétigny-sur-Orge, France; 3Biotechnology High Performance Computing Software Applications Institute, Telemedicine and Advanced Technology Research Center, U.S. Army Medical Research and Development Command, Fort Detrick, MD 21702, USA; plu@bhsai.org; 4Henry M. Jackson Foundation for the Advancement of Military Medicine, Inc., Bethesda, MD 20817, USA; 5Michael Cuccione Childhood Cancer Research Program, BC Children’s Hospital Research Institute, Vancouver, BC V5Z 4H4, Canada; 6Center for Enabling Capabilities, Walter Reed Army Institute of Research, Silver Spring, MD 20910, USA; sidhartha.chaudhury.mil@mail.mil

**Keywords:** malaria, correlate of protection, immune signature, whole sporozoites vaccine, machine learning

## Abstract

Immune correlates of protection remain elusive for most vaccines. An identified immune correlate would accelerate the down-selection of vaccine formulations by reducing the need for human pathogen challenge studies that are currently required to determine vaccine efficacy. Immunization via mosquito-delivered, radiation-attenuated *P. falciparum* sporozoites (IMRAS) is a well-established model for efficacious malaria vaccines, inducing greater than 90% sterile immunity. The current immunoprofiling study utilized samples from a clinical trial in which vaccine dosing was adjusted to achieve only 50% protection, thus enabling a comparison between protective and non-protective immune signatures. In-depth immunoprofiling was conducted by assessing a wide range of antigen-specific serological and cellular parameters and applying our newly developed computational tools, including machine learning. The computational component of the study pinpointed previously un-identified cellular T cell subsets (namely, TNFα-secreting CD8^+^CXCR3^−^CCR6^−^ T cells, IFNγ-secreting CD8^+^CCR6^+^ T cells and TNFα/FNγ-secreting CD4^+^CXCR3^−^CCR6^−^ T cells) and B cell subsets (i.e., CD19^+^CD24^hi^CD38^hi^CD69^+^ transitional B cells) as important factors predictive of protection (92% accuracy). Our study emphasizes the need for in-depth immunoprofiling and subsequent data integration with computational tools to identify immune correlates of protection. The described process of computational data analysis is applicable to other disease and vaccine models.

## 1. Introduction

Identifying immune signatures of protection induced by vaccination is a key approach to accelerate and rationalize vaccine development, to simplify immune assays that determine efficacy in clinical trials, and to better understand the impact of immune baselines on vaccine effectiveness. The immune system has evolved to be redundant, and vaccines, like prior natural infection, may protect through multiple mechanisms [1]. Immune correlates of protection, or even surrogates of protection, are still unknown for most diseases, especially when the pathogen has developed multiple immune evasion strategies, as is the case in malaria [2].

Malaria is a parasitic infection caused in humans by five *Plasmodium* species and transmitted by *Anopheles* mosquitoes. In 2019, an estimated 229 million cases occurred worldwide, and 409,000 deaths were still reported, despite the progress in a growing number of countries with a low burden of malaria towards the goal of zero malaria [3]. *P. falciparum* (*Pf*) is responsible for the highest burden of malaria worldwide due to severe clinical outcomes that affect mainly pregnant women and children under 5 years of age. The development of a protective vaccine has proven to be challenging. The most advanced vaccine, RTS,S/AS01B (Mosquirix^®^), is a subunit vaccine targeting the major surface antigen of *Pf* sporozoites (SPZ), the circumsporozoite protein (CSP). This vaccine has been part of a large-scale pilot implementation program in children to assess safety and benefits during delivery through standard public health mechanisms, and is now recommended by the World Health Organization (WHO) in regions with moderate to high transmission of *Pf* malaria. However, it has shown only modest and transient efficacy in reducing the incidence of clinical malaria by about 30% in young children [4,5]. A modified version of this vaccine (R21) elicited 77% protection in a phase 2 trial and is the first vaccine to achieve the level of protection required by the WHO [6]. In addition to recombinant and vectored malaria vaccines, it is possible to use the classic approach of whole-pathogen (here, whole sporozoite) vaccines [7]. For a whole sporozoite vaccine, protective antigens do not need to be identified and genetic restriction within the human population may be circumvented. Attenuation of the sporozoite vaccines has been accomplished either through radiation [8,9,10] or genetic attenuation by deletion of genes required to complete liver-stage development [11]. Radiation-attenuated SPZ (RAS) have long been known to induce sterile immunity [12,13,14]. Immunization with over 1000 bites of irradiated mosquitoes delivered over an extended time induces up to 93% sterile immunity against controlled human malaria infection (CHMI), occurring within 10 weeks of immunization (reviewed in [15,16]). Therefore, immunization via mosquito bites with RAS *Pf* SPZ (IMRAS) has been considered a “gold standard” malaria vaccine, even if its implementation in the field as a vaccine for large populations may be difficult despite huge advances [17]. Besides developing this platform as vaccine, the RAS model has been a valuable tool for the discovery of new pre-erythrocytic stage antigen targets [18,19,20] and for the characterization of the immune responses that confer sterile immunity in rodent models [21]. However, the immune correlates of sterile immunity appear to be different between mice and humans [16,21,22,23,24,25]. Addressing this shortfall in identifying human correlates of protection, a clinical trial was designed to achieve only 50% of sterile immunity against homologous CHMI and to generate a repository of cryopreserved samples [26]. The vaccine schedule involves the delivery of 800 to 1200 infected bites in five immunization sessions and achieved a vaccine efficacy of 55% in the first cohort (6/11), and—due to variabilities in mosquito infection rates and number of infectious bites received during vaccination in the second cohort—a vaccine efficacy of 90% in the second cohort (9/10) [26]. Sterile immunity was assessed in participants in the clinical trial by microscopy (thick smears) for the presence of blood-stage parasites. Subjects were tested twice daily for 18 days post CHMI and then every other day until day 28 post CHMI. Only individuals that remained negative by microscopy (and symptom-free) until day 28 were scored as having sterile immunity.

In the present study, we established the immunological landscape of malaria-specific adaptive immune responses by performing longitudinal analyses of cellular and serological parameters. We combined immunoprofiling with an integrative computational approach that we previously developed to establish immune signatures of clinical adjuvant formulations for recombinant vaccines [27,28,29] to define immune signatures of immunization and protection. IMRAS induced a broad range of humoral and cellular immune responses, but unexpectedly, only the functionality of the T cell response correlated significantly with protection. This finding contrasts the immune correlates reported for RTS,S that clearly demonstrated the role of antibodies in protection. In the present study, the frequency of TNFα-secreting CD4^+^CXCR3^+^ T cells responding to SPZ stimulation was significantly predictive of protection. The machine learning approach using the random forest model also identified CD19^+^CD24^hi^CD38^hi^CD69^+^ activated transitional B cells, TNFα-secreting CD8^+^CXCR3^-^CCR6^−^ T cells, IFNγ-secreting CD8^+^CCR6^+^ T cells, and TNFα/IFNγ-secreting CD4^+^CXCR3^-^CCR6^−^ T cells as parameters predictive of protection. Unexpectedly, the baseline level of TNFα-secreting CD4^+^ T cells in response to SPZ stimulation ex vivo was a significant factor that predicted protection.

## 2. Materials and Methods

### 2.1. Study Design

Samples for this study, i.e., sera and peripheral blood mononuclear cells (PBMCs), were collected under a clinical protocol (www.clinicaltrials.gov trial ID NCT01994525, accessed on 12 January 2022) from an open-label clinical study for safety and identification of biomarkers of protection in two cohorts of healthy malaria-naïve adults, who received five immunization sessions with bites from *Anopheles stephensi* mosquitoes that were either infected with *Pf*RAS (true-immunization, n = 21) or non-infected (mock-immunization, n = 5). The study procedures of immunization and CHMI to assess protection have been previously described [26]. Immunization parameters were selected for 50% protection based on prior clinical data. Leukapheresis was performed to collect plasma and peripheral blood mononuclear cells in order to identify biomarkers of protection, including host response and antigenic targets, by comparing protected and non-protected subjects.

Leukapheresis samples were available from four different time points: pre-immune (T0) as a reference sample for each volunteer; post-immune (T1), after the third immunization (pre-CHMI), as a signature of immunization; day 5–6 post-CHMI (T2) as an early time point after infectious SPZ inoculation via mosquito bites; and 3–4 months post-CHMI (T3) to look for editing of the immune response after CHMI (Figure 1).

Longitudinal samples (cryopreserved PBMCs, sera) from time points T0 to T3 were available for 12 immunized volunteers, 6 from each cohort. The distribution of protected (P)/non-protected (NP) was 3/3 in cohort 1 and 5/1 in cohort 2.

### 2.2. Enzyme-Linked Immunosorbent Assay (ELISA)

Antibodies to circumsporozoite protein (CSP), apical membrane antigen (AMA)-1, and cell traversal protein for ookinetes and sporozoite (CelTOS) were measured on day 0, 14 days post third immunization (paralleling the leukapheresis time point), the day of CHMI, and 28 days after CHMI as previously reported [26]. The cutoff for specific responses was an optical density (OD) of less than 0.1 at a 1:50 serum dilution.

### 2.3. Immunofluorescence Antibody Assay (IFA) Using SPZ

Antibody responses to whole SPZ were measured by IFA activities to SPZ pre-immunization and 22 days after the last immunization (day of CHMI). Serum antibody levels were assessed by IFA against air-dried *P. falciparum* 3D7 clone of NF54 SPZs as previously described [18]. The cutoff for a positive, specific response was no signal at a 1:50 serum dilution.

### 2.4. Electro-Chemiluminescence-Based Multiplex Assay (MSD) for Antibody Detection

The multiplex MSD methodology is based on the Mesoscale U-PLEX platform and 10-spot MSD plates (MSD, Gaithersburg, MD). Antibodies directed against 7 antigens were multiplexed: 32-mer peptide representing the CSP-repeat (NANP), epitope Pf16 within the C-terminus of CSP, gSG6-1 and gSG6-2 derived from saliva protein gSG6 as markers of exposure to *Plasmodium*-infected mosquitoes [30], erythrocytic antigen (merozoite surface protein (MSP)-1), and gametocyte antigens Pfs16 and Pfs25. This multiplexed method was described elsewhere [31] with no competition for closely related antigens. Briefly, 200 µL of each biotinylated protein (300 nM) was combined with 300 µL of a unique U-PLEX linker and incubated for 30 min at room temperature before the addition of 200 µL of stop solution. All U-PLEX coupled protein solutions were combined for the multiplexing. Fifty µL of this 1 X multiplex coating solution was added to each well of the U-PLEX 10-assay plates. After incubation, plates were washed, and 50 µL of sera were added to each well. After 1 h of incubation and 3 washing steps, addition of the detecting antibody (SULFO-TAG goat anti-human antibody) allowed the detection of a specific chemiluminescent signal with the MESO QuickPlex SQ 120 (MSD), per manufacturer’s instructions.

### 2.5. B Cell ELISpot

Human B cell responses were measured using ELISpot assay kits (Mabtech Inc, Cincinnati, OH, USA) following the manufacturer’s instructions. Thawed PBMCs were stimulated with 1 µg/mL of the toll like receptor (TLR)7/8 ligand R848 and 10 ng/mL recombinant human interleukin (IL)-2 in culture medium (RPMI-1640 containing 10% fetal bovine serum, Pen/Strep, L-glutamine, NEAA, sodium pyruvate, 2-mercaptoethanol) for 36 h. ELISpot plates were coated with recombinant PfAMA-1 (1 µg/mL), PfCSP (1 µg/mL), PfCelTOS (0.5 µg/mL), PfMSP-1 (1 µg/mL), Pf SPZ (30,000 lysed SPZ/well) [25], or anti-IgG monoclonal antibody MT91 (positive control to capture all secreted IgG). Prior to plating, B cells were purified by magnetic enrichment using CD19 microbeads (Miltenyi Biotec, San Diego, CA, USA) per manufacturer’s instructions. Enriched cells were plated at a concentration of 5 × 10^4^ cells/well. ELISpot plates were analyzed using the AID Autoimmun Diagnostika GmbH ELISpot reader (Strassberg, Germany) and software. The values of the positive assay control (MT91) were used to confirm the viability and functionality of the PBMC.

### 2.6. Flow Cytometric Analysis

Cryopreserved PBMCs from the four time points were stimulated with either (1) CSP peptide pool (15-mer peptides overlapping by 11 AA representing the CSP (3D7) vaccine antigen) at 5 µg/mL final concentration, (2) salivary gland dissected SPZ from *Anopheles stephensi* infected with NF54 *Pf* isolate (15,000 lysed SPZ/1.10^6^ PBMCs), (3) lipopolysaccharide (LPS) (B cell positive control), (4) or Cytostim (T cell positive control) or (5) media alone (negative control). Cells were cultured for 16 h (or 24 h for stimulation with SPZ lysates) (37 °C, 5% CO_2_) in complete medium (RPMI-1640 (Life Technologies, Waltham, MA, USA) containing 10% human serum (Gemini Bio-Products, West Sacramento, CA, USA)) at a concentration of 1 × 10^7^ cells/mL. For cytokine analysis, Golgi Plug (3 µL/mL) was added during the last 14 h of stimulation. Following antigen stimulation, PBMCs were washed and stained with anti-human CD69-biotin (clone REA824) for 15 min at 4 °C in FACS solution (0.5% human serum and 0.1% sodium azide in PBS) (T cell panel). Cells were further incubated with anti-biotin microbeads (Ultrapure, Miltenyi Biotec) for 15 min at 4 °C. After washing, an antibody cocktail with fluorochrome-conjugated antibodies against CD4-APC H7 (clone RPA-T4), CD8-FITC (REA715), CD185-PEVio770 (clone REA103), CD183-BV711 (clone 1C6/CXCR3), CCR6-APCR700 (clone 11A9), and Biotin-PE (clone REA 746) and L/D fixable blue dead cell stain kit (Thermo Fisher Scientific, Waltham, MA, USA) was added and incubated for 45 min at 4 °C. For intracellular staining (ICS), fixation/permeabilization was performed with Foxp3/transcription factor staining buffer set (eBioscience, San Diego, CA, USA) for 45 min at 4 °C in the dark. After washing, an intracellular antibody cocktail with fluorochrome-conjugated antibodies against CD3-PerCP-Cy5.5 (clone SK7), Foxp3-PECF594 (clone 259D/C7), tumor necrosis factor (TNF)α-V450 (clone MAb11) and interferon (IFN)γ-APC (clone 4SB3) was added and incubated for 45 min at 4 °C. For the B cell panel, cells were washed after stimulation, and an antibody cocktail with fluorochrome-conjugated antibodies against CD3-Vioblue (clone BW264/56), CD19-PerCP-Vio (clone LT19), CD20-APC-Vio (clone REA780), CD24-PE (clone REA832), IgD-FITC (clone IgD26), CD27-PE Vio700 (clone REA499), CD69-BV711 (clone FN50), and CD38-APC (clone REA671) and L/D fixable blue dead cell stain kit (Thermo Fisher) was added and incubated for 45 min at 4 °C. Cells were enriched when necessary (T cells) and acquired on a BD LSR Fortessa. Cell viability of thawed PBMCs was >92% as measured by a Luna-FL™ Dual Fluorescence cell counter (fluorescence protocol with acridine orange/propidium iodide (AO/PI) to determine cell viability). Viability of the cells after overnight stimulation and staining was >84%. For the T cell panel, lymphocytes were first gated based on Forward- (FSC) vs. Sideward Scatter (SSC), then single cells, followed by viability, and the lineage marker CD3 (Appendix A). Antigen-specific cells were gated based on co-expression of CD4 or CD8 with CD69. Subsequent gating for CXCR5 and Foxp3 expression differentiated circulating follicular T-helper cells (cTfh) (CXCR5^+^Foxp3^-^) and follicular regulatory T cells (Tfr) (CXCR5^+^Foxp3^+^) in CD4^+^ T cells. Tfh subsets were based on the expression of CCR6 and CXCR3 to identify Tfh1, Tfh2, and Tfh17 cells. Subsequent gating of CD4^+^CD69^+^ and CD8^+^CD69^+^ was also performed based on CCR6 and CXCR3 expression. Finally, the frequency of IFNγ- and TNFα-secreting T cells was determined for each subset. For the B cell panel, lymphocytes were first gated based on Forward- vs. Sideward Scatter, then single cells, followed by viability and the lineage marker CD19 (Appendix A). Naïve and memory B cell subsets were gated on CD27 expression. Then, memory B cells (MBC) were further gated on IgD expression (switched IgD^−^ and un-switched IgD^+^ subsets) and CD20^-^CD38^+^ (plasmablasts). Transitional B cells were defined by co-expression of CD24^hi^ and CD38^hi^. Finally, CD69 expression for each subset determined the reactive compartment to antigen stimulation. Flow cytometric data were analyzed using FlowJo V9 (Treestar, Ashland, OR, USA). Raw data were used for the statistical analyses, rather than data after subtracting the values from media-stimulated (background)-controls, because we hypothesize that activated and specific cells can be found in unstimulated PBMCs due to the short time after the third immunization for leukapheresis.

### 2.7. MSD for Cytokine Detection

Cryopreserved PBMCs from each study subject from time points T0 (pre-immune) and T1 (pre-CHMI) were stimulated with AMA-1, CelTOS, CSP, MSP-1, and thrombospondin related adhesion protein (TRAP) peptide pool, high (135,000 SPZ/mL) or low (45,000 SPZ/mL) concentration of SPZ lysates, media alone (control stimulation), or anti-CD3. Cell viability of thawed PBMC was >92% as measured by a Luna-FL™ Dual Fluorescence cell counter (fluorescence protocol with AO/PI to determine cell viability). The Mesoscale Discovery’s 10-plex human pro-inflammatory panel kit (IL1β, IL8, IL2, IL4, IL6, IL10, IL12p70, IL13, IFNγ, TNFα) was used to analyze culture supernatants according to the manufacturer’s protocol. Plates were read on a QuickPlex SQ120 (Mesoscale, Gaithersburg, MD, USA). Cytokine levels induced by anti-CD3 stimulation fell within the range of more recently cryopreserved as well as freshly obtained PBMC, indicating that long-term storage did not impact the functionality of the cells.

### 2.8. Statistical Analysis

We integrated the data obtained from all immunoassays for each subject and carried out univariate analysis, comparing each immune response to its pre-immune reference point, at the group level, to identify the subset of immune responses that were immunization-induced. With those data, we carried out a univariate analysis between subjects from cohorts 1 and 2 to identify differences between cohorts, and between protected and non-protected subjects to identify correlates of protection. Finally, we used machine learning to determine how well we could distinguish protected and non-protected subjects, and what multi-factorial combination of immune factors was most responsible for making this distinction. A flowchart outlining the data analysis strategy is shown in Appendix A.

### 2.9. Univariate Analysis

To determine which immune responses showed immunization-induced changes, univariate analysis for each immune measure was carried out, comparing pre-immune vs. post-immune responses. If normally distributed, as determined by Shapiro–Wilk tests, paired Student’s t-tests were used to calculate statistical significance. If not normally distributed, the Wilcoxon signed-rank test was applied. After calculating *p*-values for immune parameters in the data sets, a correction for multiple comparisons was made (resulting in a corrected *p*-value) using the Benjamini–Hochberg correction. Immune measures in which comparison to the pre-immune data showed a significant difference at *p* < 0.05 and a false discovery rate of q < 0.20 were selected for subsequent analyses. Then, a more stringent set of criteria was applied to define IMRAS immune signature (*p* < 0.05, q < 0.05). A similar pipeline was used to compare cohorts and protection status, with a significant difference at *p* < 0.05 and a false discovery rate of q < 0.05.

### 2.10. Multivariate Analysis and Machine Learning

Correlation matrices for the data set were generated by calculating the Spearman correlation coefficient between all immune measures. Spearman’s ρ statistic was used to calculate *p*-values for each correlation estimate. Only correlation coefficients with *p* < 0.05 were retained for further analysis to ensure that only high-confidence correlations were used in subsequent analyses; all others were set to ‘0’. Hierarchical clustering (R package *hclust* function) was carried out with the R statistical software [32] using the *hclust* function to group correlated immune measures and to define immune clusters based on a cutoff criterion of having a correlation coefficient of at least 0.40, using the *cutree* function.

For the random forest model (based on the R *caret* package), we used all immunization-induced immune factors (61 parameters measured at T0 and at T1) to predict protection for all subjects in the study (complete dataset for n = 12). The model was trained using the repeated *cv* method, subsampling the data set by 5-fold and resampling 100 times. The *varImp* function was used to determine the variable importance for each generated model, and the average variable importance across all models was reported to assess the relative importance of each immunization-induced immune factors in predicting the protection status. The *mtry* parameter corresponded to the number of variables available for splitting at each tree node. The accuracy of the model was the percentage of correctly classified individuals. The kappa value (κ) was normalized at the baseline of random chance on the dataset and was of particular interest for an imbalanced model, as is the case in this study. All R analysis scripts used in this study are publicly available at https://github.com/BHSAI/IMRAS (accessed on 12 January 2022).

## 3. Results

### 3.1. Immunoprofiling of IMRAS-Induced Humoral and Cellular Immune Response

We established the immunological landscape of responses induced by IMRAS by conducting a wide range of cellular and serological assays (Table 1) on longitudinal PBMC (12 subjects) and serum samples (16 subjects) to identify immune correlates of vaccination and protection.

The wide range of collected immune measures consisted of: (i) serological responses to SPZ and 10 different SPZ antigens using immunofluorescence assays (IFA), enzyme-linked immunosorbent assay (ELISA), and a multiplexed antigen testing platform (MSD) [31]; (ii) frequency of IgG-secreting B cells specific to AMA-1, CelTOS, CSP, MSP, or SPZ; (iii) functional phenotype of activated (CD69^+^) B cells (i.e., naïve, memory, memory switched IgD^−^ and un-switched IgD^+^, transitional CD19^+^CD24^hi^CD38^hi^ B cells and plasma blasts) in response to SPZ stimulation; and (iv) functional phenotype of circulating follicular helper T cells (cTfh1, cTfh2, and cTfh17) after SPZ and CSP stimulation. CXCR5^+^ T cells are essential for B cell maturation in the germinal center [33]. To evaluate the cellular response, we measured the phenotype of CD4^+^ T cells and CD8^+^ T cells (expression of CXCR3 and CCR6, which promote tissue migration of T cells) [34,35] by flow cytometry after SPZ and CSP stimulation. The functionality of these cells was addressed by intracellular staining (ICS) of IFNγ and TNFα. To assess Ag-specificity by flow cytometry, we used the CD69^+^ activation marker because it is widely expressed on various lymphocyte populations after activation (CD4^+^ and CD8^+^ T cells as well as B cells) and rapidly upregulated (<2 h) [36,37]. The gating strategies are detailed in Appendix A. Ten cytokines (IL1β, IL2, IL4, IL6, IL8, IL10, IL12p70, IL13, IFNγ, TNFα) were measured in the supernatant of PBMC cultures stimulated with different malarial antigens (AMA-1, CelTOS, CSP, SPZ, TRAP).

By employing a statistical data integration method that incorporated all the immune data from both cohort 1 and 2 at T0 and T1, we isolated 61 out of 1316 immune measures that were significantly different from baseline (*p* < 0.05, q < 0.2) (Appendix A). We then used this selected dataset for subsequent analysis in order to decipher differences in cohorts and protection status. Among these 61 immune measures, we focused on 26 measures that were significantly different after immunization with more stringent criteria (*p* < 0.05, q < 0.05) to define the IMRAS-associated immune signature. We carried out a correlation analysis with hierarchical clustering on these selected measures to identify immune parameters that significantly correlated (Spearman, ρ > 0.4, *p* < 0,05) (Figure 2). The correlation matrix revealed strong positive correlations between the SPZ-specific B cell populations and CSP- and SPZ-specific T cell responses. Interestingly, strong negative correlations were observed between IL-8 produced by antigen-specific PBMC and B cell populations.

### 3.2. IMRAS Humoral Response Is Mainly Driven by CSP

IMRAS induced a strong humoral response, as assessed by antibody titers against SPZ (IFA, *p* < 0.001, q = 0.01, n = 16) (Figure 3A), AMA-1 (ELISA, *p* = 0.002, q = 0.03), full CSP (ELISA, *p* < 0.001, q = 0.01) (Figure 3B) and CSP-NANP (MSD, *p* < 0.001, q = 0.02) (Figure 3C). Antibody titers against AMA-1 measured by ELISA were not induced in all volunteers (Figure 3B), and these non-responding individuals (n = 3) remained seronegative even after CHMI. Antibody titers against the CSP C-terminus, a merozoite surface protein (MSP-1) epitope, two mosquito saliva peptides from salivary gland protein gSG6, and two early gametocyte antigens (Pfs16/Pfs25) were not significantly induced after immunization (Figure 3C). Stratifying the responses to AMA-1 and CSP by protection status demonstrated that antibody levels to AMA-1 were lower in protected individuals after CHMI (T3, *p* = 0.021, q = 0.15), while the antibody titers to CSP were higher in protected individuals prior to and after CHMI (T2, *p* = 0.042, q = 0.15; T3, *p* = 0.037, q = 0.15).

### 3.3. IMRAS Vaccination Leads to Significant Expansion of B Cells Specific for a Range of Plasmodial Antigens

Changes in antigen-specific B cells were assessed first by ELISpot due to the high sensitivity and the ability to screen a range of plasmodial antigens. The frequency of SPZ-specific IgG-secreting B cells (*p* < 0.001, q = 0.01, n = 10/13) and CSP-specific IgG^+^ B cells (*p* < 0.001, q = 0.004, n = 11/16) increased after vaccination in most subjects (Figure 4A). Half of the subjects (8/16) had IgG^+^ Memory B cells (MBC) specific to AMA-1 (*p* = 0.002, q = 0.03) and MSP (*p* = 0.001, q = 0.03). These responses were also identified in the correlation matrix (Figure 2). Next, we determined the functional phenotype of antigen-specific (CD69^+^) B cells specific to SPZ (n = 12) (Figure 4B). The following B cell subsets showed significant increases in frequency in peripheral blood: (1) CD19^+^CD27^−^ B cells (*p* < 0.001, q = 0.002), (2) switched (*p* < 0.001, q = 0.007), and (3) unswitched (*p* < 0.001, q = 0.004) MBC) subpopulations after immunization (T1) as well as CD38^hi^CD24^hi^ transitional B cells (*p* = 0.001, q = 0.03), but no significant increase in circulating plasmablasts after immunization. The latter observation is not surprising given that the collection time point after vaccination was not optimal for detecting the transient presence of the cells in peripheral blood.

### 3.4. IMRAS Induces Significant Levels of CSP-Specific Circulating Follicular T Helper Cells

Follicular helper T cells (Tfh) are important cells for B cell maturation in germinal centers. Since these tissue resident cells are typically not accessible when working with peripheral blood and human donors, we measured the circulating counterpart [28,38,39,40] of CD4^+^ T cells expressing CXCR5 (cTfh) and their functional subsets based on the expression of the chemokine receptors CCR6 and CXCR3: cTfh1 are CXCR3^+^ and CCR6^−^, cTfh17 are CXCR3^−^ and CCR6^+^, and cTfh2 are CXCR3^−^ and CCR6^−^ (Appendix A). In addition, we and others observed one cTfh subset that has not been characterized yet: CD4^+^ CXCR5^+^CXCR3^+^CCR6^+^ [33,38,41]. After CSP stimulation, a significant decrease in the relative frequency of these CCR6^+^CXCR3^+^ cTfh cells among CD69^+^ cTfh cells (*p* = 0.003, q = 0.04) was associated with an increase in the relative frequency of cTfh2 subtype (Figure 5A). The same trend, albeit not statistically significant, was observed after SPZ stimulation.

When testing for correlations between the various humoral measures (Figure 2), three distinct clusters corresponded to specific readout parameters (ELISA/MSD, ELISpot, flow cytometry). However, correlations were also found between AMA1 ELISA and SPZ-specific B cell ELISpot (Spearman, ρ = 0.54, *p* = 0.03), and between SPZ-MBC and SPZ-IFA (Spearman, ρ = 0.66, *p* = 0.02).

### 3.5. IMRAS Circulating Cellular Response Is Driven by IFNγ- and/or TNFα-Secreting CD4^+^ T Cells

Stimulation with either SPZ or CSP peptides significantly induced TNFα- and TNFα/IFNγ-secreting CD4^+^ T cells. These functional CD4^+^ T cell subsets differed significantly in their expression of the CXCR3^+^ homing receptor after SPZ (*p* = 0.02, q = 0.03) (Figure 5B) or CSP (*p* = 0.001, q = 0.02) stimulation (Figure 5C), but not in the expression of CCR6. No significant differences in the phenotype (i.e., expression of CXCR3, CCR6) or functional phenotype (intracellular cytokine) of antigen-specific, circulating CD8^+^ T cells were observed after either antigen stimulation.

### 3.6. Associations between Immune Measures and Protection Using Univariate Analysis

Performing simple statistical analysis between antibody responses at T1 and protection did not reveal any correlations. In contrast, cellular immune measures were able to distinguish samples from protected vs. non-protected individuals: the frequency of antigen-specific CD4^+^T cells producing TNFα after in vitro stimulation with native antigens (SPZ lysates) was significantly higher in protected individuals (Figure 6). Interestingly, this difference between protected vs. non-protected individuals could be detected in pre-immune samples as well as in pre-CHMI samples. This difference was seen when gating only on antigen-specific CD69^+^CD4^+^T cells (T0, *p* = 0.003, q = 0.037; T1, *p* = 0.007, q = 0.06) or on CD69^+^CD4^+^CXCR3^+^ T cells (*p* = 0.001, q = 0.023) (Figure 6).

Protected individuals had a higher frequency of TNFα-secreting CD4^+^ T cells than non-protected volunteers at the pre-immunization time point. Immunization led to an expansion of this subset in both groups, but the difference in frequencies between protected and non-protected volunteers was maintained (*p* = 0.007, q = 0.06).

Within antigen-specific CD8^+^ T cells, an association between TNFα-producing cells and protection was observed in CD8^+^CXCR3^-^CCR6^-^ T cells after stimulation with CSP, but with less stringent criteria (*p* < 0.05, q < 0.2). Similarly to the CD4^+^ compartment, these protection-associated differences were observed at the pre-immune (*p* = 0.03, q = 0.19) and the pre-CHMI time points (*p* = 0.02, q = 0.19). Interestingly, no differences between protected vs. non-protected individuals were observed after stimulation with SPZ. This may be due to the timing of the in vitro stimulation, as SPZ would require cross-presentation of epitopes after antigen processing, and this may not be efficient after 18 h of stimulation compared to the stimulation with CSP-peptides.

The analysis of the cytokine profile of PBMC of the longitudinal samples after stimulation with SPZ, CelTOS, and TRAP also revealed significant differences associated with the protection status (Appendix A). The cytokine profile was established by generating correlation matrices (Spearman method). Interestingly, there was no significant difference between protected vs. non-protected individuals at T0 as had been observed when performing intracellular staining of T cell subsets.

### 3.7. Predictive Immune Model Using Machine Learning

To determine which immune features were most predictive of protection, we applied a machine learning approach using the random forest method. We focused the analysis on T0 and T1 immune measures previously selected to assess protective immune signatures of protection. By running the random forest model on the 61 immune measures that were modified after immunization from baseline (*p* < 0.05, q < 0.2) (Appendix A), we reached only 65% accuracy at T1 (κ = 0.07, *mtry* of 21 parameters) with a selective bias toward protected subjects, as the data set was imbalanced (67% of volunteers were protected). We determined the relative importance of each parameter in the random forest model and selected the following six first parameters for subsequent analysis: TNFα-secreting CD4^+^ T cells after SPZ stimulation, CSP-specific TNFα-secreting CD8^+^CXCR3^−^CCR6^−^ T cells, SPZ-specific CD19^+^CD69^+^CD38^hi^CD24^hi^ transitional B cells, SPZ-specific TNFα-secreting CD4^+^CXCR3^+^ T cells, CSP-specific IFNγ-secreting CD8^+^CCR6^+^ T cells, and CSP-specific TNFα- and IFNγ-secreting CD4^+^CXCR3^−^CCR6^−^ T cells. By adding the two significant baseline parameters (SPZ-specific TNFα-secreting CD4^+^ T cells and CSP-specific TNFα-secreting CD8^+^CXCR3^−^CCR6^−^ T cells at T0) to these six parameters, the predictive model achieved 88% accuracy (κ = 0.64). Two out of four non-protective individuals (error rate of 0.5) and 8 out of 8 protected volunteers (no error rate) were correctly assigned. The relative importance of each immune parameter is summarized in Table 2.

Principal component analysis (PCA) with these eight parameters (Figure 7) showed the direction of each parameter in discriminating protected vs. non-protected individuals. In conclusion, in addition to antigen-specific TNFα-secreting CD4^+^ T cells identified by univariate analysis, the machine learning approach using the random forest model also identified the CD69^+^CD19^+^CD24^hi^CD38^hi^ activated transitional B cell subtype, antigen-specific TNFα-secreting CD8^+^CXCR3^−^CCR6^−^ T cells, IFNγ-secreting CD8^+^CCR6^+^ T cells, and TNFα/IFNγ-secreting CD4^+^CXCR3^−^CCR6^−^ T cells as important factors that are predictive of protection.

## 4. Discussion

The present study pursued two distinct goals. First, we sought to determine whether an integrative approach combining immunoprofiling and machine learning, originally developed by our team to assess recombinant vaccine platforms and adjuvant formulations, could be applied to an entirely different vaccine platform, thus establishing its wider applicability. Secondly, if the approach was successful, it would allow us to determine the predictive immune signature of IMRAS immunization. Indeed, we succeeded in identifying a combination of immune parameters that predict the protection status prior to CHMI with an accuracy of 88% despite the limited size of the clinical cohorts. Distinctly different from recombinant malaria protein-based vaccines, the most predictive immune parameters of protection were specific cellular responses. An unexpected component of the protective immune profile, however, was the vaccinees’ pre-immune baseline response to malaria antigens.

Numerous attempts have previously been made to identify correlates of protection following vaccination against various pathogens, but even trials with large cohorts have failed to pinpoint specific immune measures that predict vaccine efficacy. While the sample size of our study was relatively small, it still succeeded in identifying immune correlates of protection by combining immunoprofiling and machine learning. Our study results suggest that in-depth immunoprofiling is a necessary step to define the immune signatures associated with a particular vaccine formulation or diseases outcome. Once the profiles are established, multi-variate analyses and machine learning enable the identification of parameters that drive a specific profile and, thus, identify correlates of protection or surrogate markers of vaccine efficacy.

Our data demonstrate that the IMRAS vaccine induces broad, high-titer antibody responses to a variety of SPZ antigens. Our study could not find a correlation between the magnitude of antibody responses and protection. As has been shown for other infectious diseases, quality rather than quantity of the humoral immune response may be more indicative of protective immunity [42,43]. Reports from various clinical trials demonstrate the functionality of antibodies induced by irradiated sporozoites (reviewed in [15]). However, the ability of the various functional assays to predict protection against disease is still being investigated [15,44]. Alternatively, neutralizing activity against the sporozoites could be mediated by antibodies specific to antigens other than CSP, i.e., CelTOS, TRAP, MSP-1, and AMA-1 [45].

At the cellular level, the vaccine leads to the induction and activation of antigen-specific naïve and memory B cells and antigen-specific cTfh2 T cell subsets that have been shown to support B cell activation, B cell maturation, and antibody production [23,33,46]. Surprisingly, none of these parameters when measured after immunization (pre-CHMI) predicted the protection status. Antibody titers correlated with protection after CHMI only, with the predictable induction of AMA-1 response in non-protected volunteers due to the presence of blood stage parasites. CSP antibody titers discriminated protected and non-protected volunteers after CHMI, in accordance with previous observations in the literature from RTS,S phase 2 clinical trials [47]. Interestingly, the frequency of circulating lymphocytes diminished after immunization. Previous reports have revealed the presence of “natural” T cells in malaria-naïve individuals that quickly proliferate when exposed to malarial antigens [48]. Vaccination with the IMRAS vaccine would induce these cells and then likely prime them to migrate to lymphoid organs and tissues and therefore, at least transiently, decrease their frequency in peripheral blood. The relative frequency of cTfh2 increases after immunization, and this subtype is known for its critical role in providing B cell help [33]. In contrast, cTfh1 cells are preferentially activated during natural infection with a negative impact on B cell maturation [49]. An increase in cTfh1 cells was observed when the RTS,S/AS01B vaccine was used in a heterologous prime-boost regimen with viral-vectored vaccines (MVA or ChAd), and negatively correlated with antibody responses [35]. In our study, cTfh1 cells that express CXCR3 and general Tfh (CD4^+^CXCR5^+^) T cells expressing both CXCR3 and CCR6 were less abundant after immunization. Since CXCR3 and CCR6 are tissue homing receptors [50,51,52], especially for the liver [53,54], these cells may have been directed to the liver considering the liver-tropism of malaria SPZ. There are no data about the role of CD4^+^CXCR5^+^CCR6^+^CXCR3^+^ T cells in B cell maturation, but these cells were already observed in other studies [33,38,41].

IMRAS cellular responses measured in the blood are driven by poly-functional T cells secreting TNFα and IFNγ after SPZ and CSP stimulations. Previous studies in mice and non-human primates have shown that a cellular immune response is likely sufficient to mediate durable protection induced by RAS [55,56,57]. IFNγ, cytotoxic CD8^+^T cells, and poly-functional CD4^+^ T cells were associated with protection [58,59,60,61]. In our study, using IMRAS, protection correlated with TNFα-secreting CD4^+^ T cells after SPZ stimulation, but unexpectedly, there was no concurrent increase in TNFα concentration in the culture supernatant, suggesting a local action of the cytokine in cell-to-cell communication. TNFα is known to help control parasite expansion at early stages of infection but acting in a context-dependent manner to increase the risk of severe malaria during natural infection, with some reports suggesting that the TNFα response capacity in malaria is controlled by TNFα polymorphism [62]. The inhibitory effect of TNFα at the liver stage has been described as indirect through the induction of yet undefined mediators secreted by hepatocytes [63]. CSP stimulation induced IFNγ-secreting as well as IFNγ- and TNFα-secreting CD4 T cells, but none of them significantly correlated with protection. Using machine learning, the random forest model approach identified the CD19^+^CD24^hi^CD38^hi^CD69^+^ transitional B cell subtype, TNFα-secreting CD8^+^CXCR3^−^CCR6^−^ T cells, IFNγ-secreting CD8^+^CCR6^+^ T cells, and TNFα/IFNγ-secreting CD4^+^CXCR3^−^CCR6^−^ T cells as important variables that predict the protection status. The relative frequency of activated transitional B cells, defined as CD19^+^CD38^hi^CD24^hi^CD69^+^ B cells, was higher in non-protected volunteers. This cell population is known to exert regulatory functions by IL-10 secretion that inhibits pro-inflammatory cytokine production by CD4^+^ T cells [64]. The frequency of circulating Ag-specific CD8^+^ T cells should be low, since immunization-primed cytotoxic T cells would home to the tissues establishing a specific tissue-resident memory compartment of cells that stay in the liver [65,66]. The chemokine receptors CXCR3 [67], CXCR6 [54,68], and CCR6 [54] and integrins (αEβ7, αLβ2) [69,70] have been implicated in T cell homing to the liver, either for the recruitment or the positioning of the cells. In our study, circulating CD8^+^ T cells that secrete TNFα and/or IFNγ mainly expressed CCR6 receptor, with or without co-expression of CXCR3. The chemokine axis CCR6^−^CCL20 is involved in inflamed tissues and has already been associated with skin homing [71]. Since the IMRAS vaccine is delivered via a thousand mosquito bites, this vaccination route could favor skin homing, as compared to PfSPZ administration via the intravenous route where the parasite load is routed directly to the liver. The target of RAS-induced protective cell-mediated immunity has not yet been clearly delineated in the literature [72], even if CSP and TRAP are confirmed targets of protective immunity in rodent models [24,73,74]. Our study also did not identify conclusive antigenic targets associated with whole SPZ vaccines. In culture supernatants, the cytokine profile after SPZ, TRAP, and CelTOS-stimulation was clearly different between protected and non-protected volunteers. The cytokine profile is driven by Th1 cytokines following SPZ stimulation with a low dose, while the cytokine profile following a higher dose of SPZ or CSP-stimulation is associated with a significant upregulation of IL4 and IL13. The importance of IL4 in IMRAS was described previously, with protection status coinciding with the expansion of parasitized red blood cell-reactive IL4-secreting memory CD4^+^ T cells [23].

Our study highlights the importance of the immune baseline as a predictor of post-vaccination responses. Before vaccination, the frequencies of TNFα-secreting T cells in response to malaria antigens were significantly different at baseline between protected and non-protected volunteers. By using an influenza vaccine as a model, Tsang et al. showed that most of the measured parameters that contribute to prediction of post-vaccination antibody responses are temporally stable baseline differences between individuals [75]. In our case, all volunteers were malaria-naïve, and no previous exposure to malaria antigens could explain the observed differences. TNFα response capacity in malaria may be controlled by TNFα polymorphism [62], and our findings provide the rationale for conducting genetic analyses of the TNF gene locus in future IMRAS vaccine trials.

Lastly, we would like to note that the study was based on a limited sample set and requires validation with larger clinical studies. The current study serves as a template on how to conduct immunoprofiling and apply computational tools including machine learning to identify immune correlates.

## 5. Conclusions

The present study revealed key signatures of the adaptive immune response to IMRAS employing an original approach that combined in-depth immunoprofiling with machine learning. Particularly, TNFα-secreting CD4^+^ T cells in response to malaria antigen stimulation, at baseline and after immunization, were associated with protection. The activation of a regulatory B cell compartment in non-protected volunteers highlights the important balance between pro-inflammatory and regulatory signals. The role of the innate immune system in shaping this adaptive response is an essential next step to better understand and further refine correlates that will provide lessons for the improvement of malaria vaccines.

## Figures and Tables

**Figure 1 vaccines-10-00124-f001:**
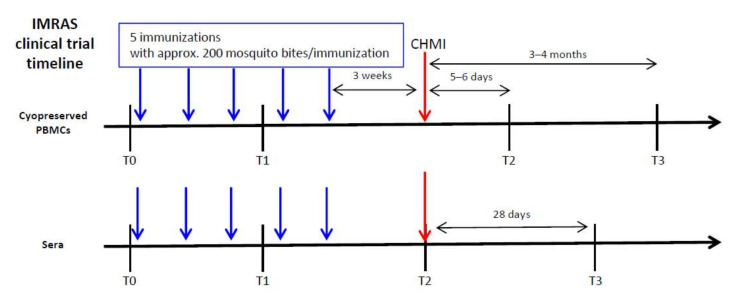
IMRAS clinical trial timeline. Cryopreserved PBMCs were available from baseline (T0), 2 weeks after the third immunization (T1), 5–6 days after controlled human malaria infection (CHMI, T2) and 3–4 months after CHMI (T3). Sera were available from baseline (T0), 2 weeks after the third immunization (T1), the day of CHMI (T2), and 28 days after CHMI (T3). The vaccine schedule included 5 immunizations with approx. 200 bites from mosquitoes infected with irradiated SPZ per immunization.

**Figure 2 vaccines-10-00124-f002:**
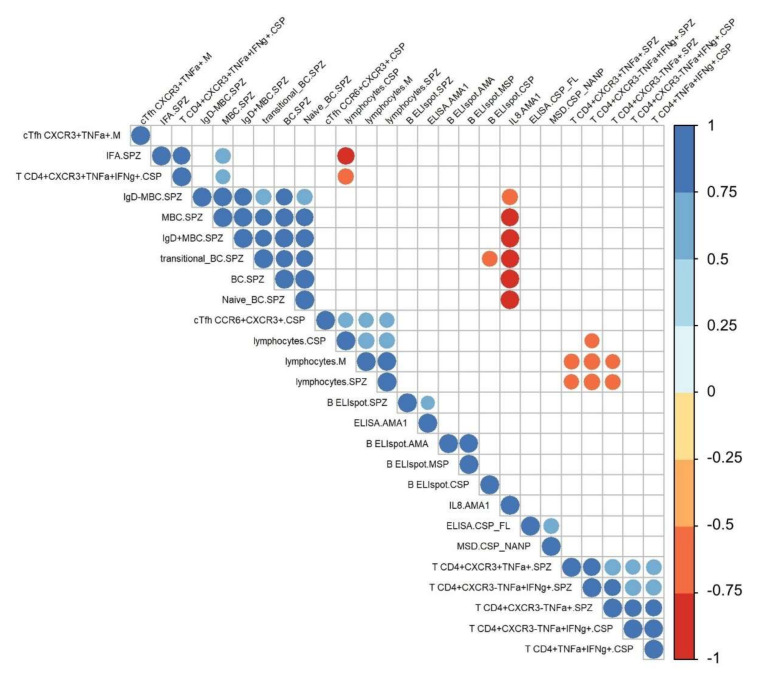
Immune signatures induced by IMRAS vaccination. The immunological landscape was established by integrating the results of all immune assays performed on longitudinal samples from IMRAS-vaccinated subjects. Twenty-six immune measures were significantly (*p* < 0.05, q < 0.05) associated with IMRAS immunization at T1. The correlation matrix of these measures shows only significant correlations (Spearman, ρ > 0.4, *p* < 0,05). The color and size of the dots (scale next to graph) indicate the degree of correlation between the different parameters (small to large indicating low to high correlation).

**Figure 3 vaccines-10-00124-f003:**
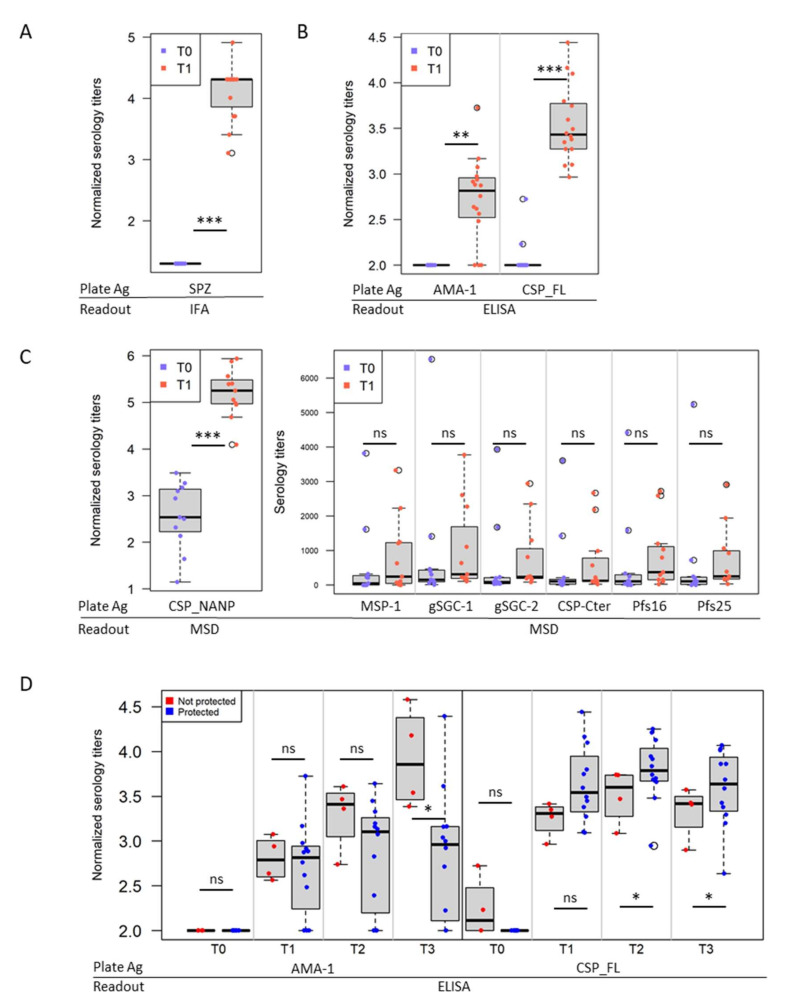
Antibody and B cell responses induced by IMRAS immunization. (**A**) Normalized log-10 transformed serology titers at baseline (T0) and after immunization (T1) specific to SPZ. Titers were assessed by IFA. (**B**) Normalized log-10 transformed serology titers at baseline (T0) and after immunization (T1) specific to AMA-1 and CSP full length (CSP_FL). Titers were assessed by ELISA. (**C**) Mean luminescence signal (measured by MSD) against MSP-1, saliva protein gSG6 (gSG6 peptides 1 and 2), CSP C-terminus (Pf16), CSP repeat (NANP), and early gametocyte antigens (Pfs16/Pfs25). (**D**) Stratification of antibody titers by protection status. Normalized log-10 transformed serology titers at baseline (T0), after the third immunization (T1), the day of CHMI (T2), and 28 days after CHMI (T3) specific to AMA-1 and CSP full length (CSP_FL) assessed by ELISA. ns = not significant, * *p* < 0.05, ** *p* < 0.01, *** *p* < 0.001.

**Figure 4 vaccines-10-00124-f004:**
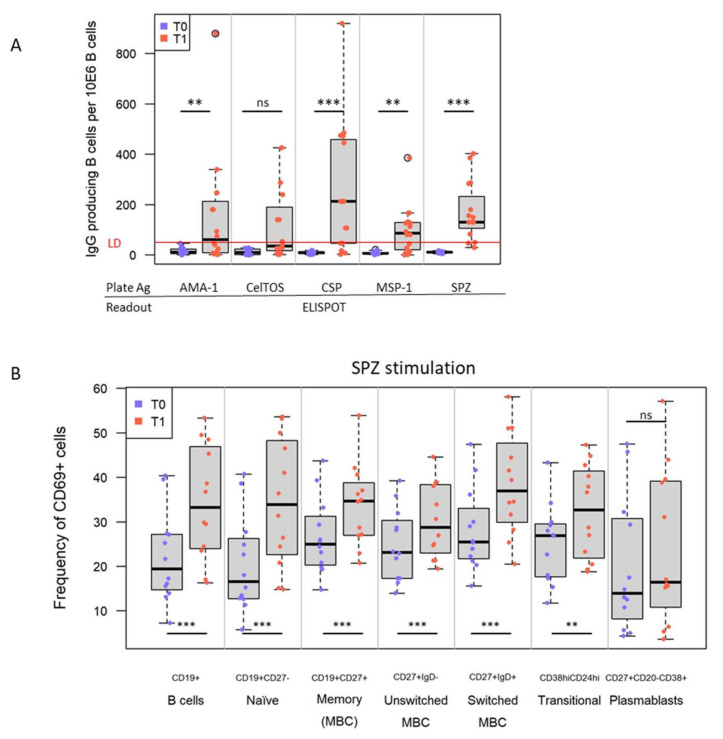
Frequency and composition of *Plasmodium*-specific reactive B cells before and after IMRAS immunization. (**A**) Antigen specificity (SPZ, CSP, CelTOS, AMA-1, and MSP-1) and frequency of *Plasmodium*-specific, IgG-producing B cells per million PBMC assessed by ELISpot. Box plots represent n = 16 subjects per time point. (**B**) Changes in the frequency and composition of SPZ-specific B cell compartment: naïve B cells, memory (MBC), switched MBC, un-switched MBC, transitional B cells, and plasmablasts (phenotypic markers indicated on *X*-axis) assessed by flow cytometry. Box plots represent n = 12 subjects per time point. ns = not significant, ** *p* < 0.01, *** *p* < 0.001. LD = limit of detection (threshold).

**Figure 5 vaccines-10-00124-f005:**
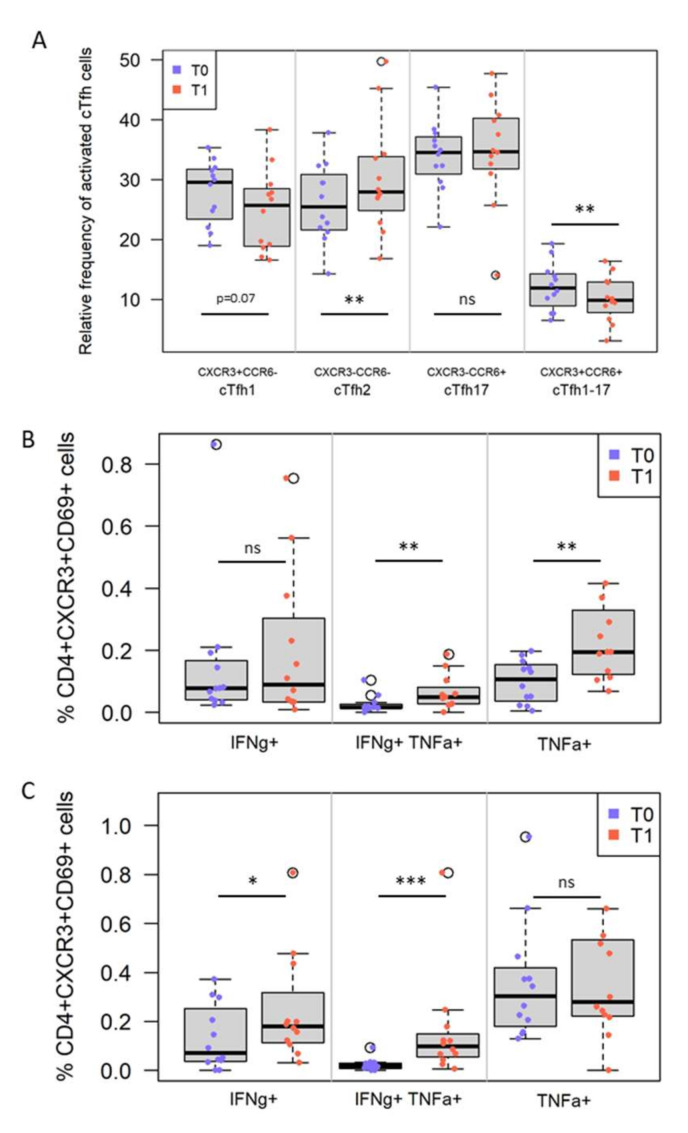
IMRAS vaccination induces significant differences in the frequency of antigen-specific T cell subsets. (**A**) Changes in the frequency and compositions of CSP-specific cTfh subsets after vaccination. Frequency of antigen-specific cTfh subsets (indicated on X-axis based on expression of chemokine receptors CXCR3 vs. CCR6) at baseline (pre-immune, T0 = blue dots) and pre-CHMI (T1 = red dots). Changes in the frequency of antigen-specific CD4^+^CXCR3^+^ T cells secreting IFNγ and/or TNFα after SPZ (**B**) or CSP (**C**) stimulation before (T0) and after immunization (T1). ns = not significant, * *p* < 0.05, ** *p* < 0.01, *** *p* < 0.001.

**Figure 6 vaccines-10-00124-f006:**
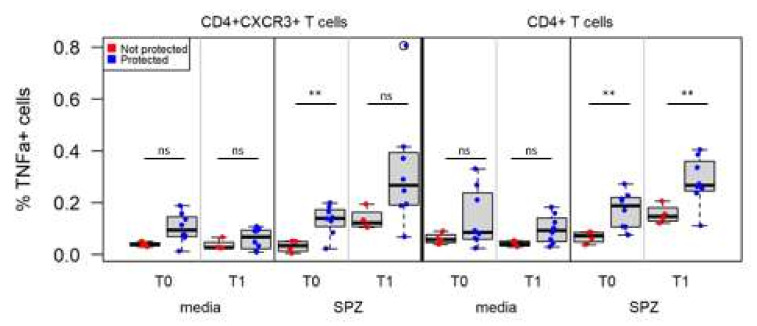
Frequencies of SPZ-specific CD4^+^ and CD4^+^CXCR3^+^ T cells are significantly higher in protected individuals. Changes in the frequency of CD69^+^CD4^+^CXCR3^+^ (Panel A) and bulk CD69^+^CD4^+^ in protected (blue) and non-protected (red) individuals at pre-immune (T0) vs. pre-CHMI (T1) time points. ns = not significant, ** *p* < 0.01.

**Figure 7 vaccines-10-00124-f007:**
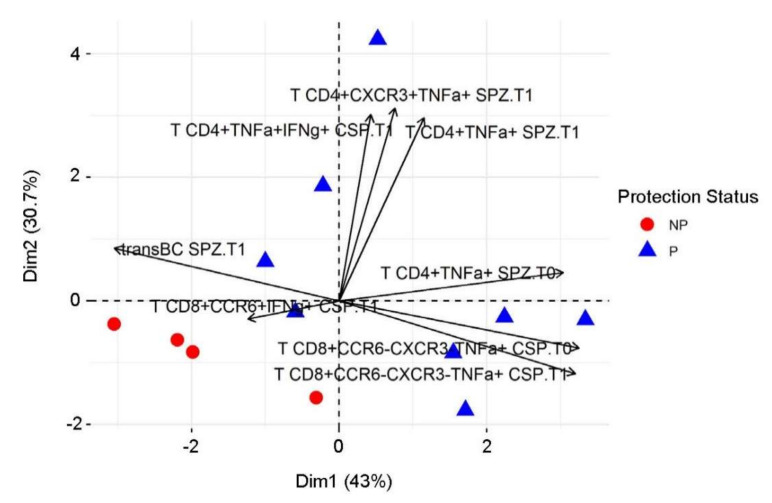
Principal component analysis of protection. Twelve individuals with a complete dataset were plotted on the first (Dim1) and second (Dim2) principal components of a principal component analysis. The color and shape of the dots (which represent individual samples) indicate protection status.

**Table 1 vaccines-10-00124-t001:** Summary of all immune measures collected to establish immune signatures correlating with sterile immunity.

Immune Parameter	Technique	Measurement	Ag-Specificity	Time-Point
Antibodies	ELISA	IgG titre	AMA-1, CSP-FL	T0, T1, T2, T3
	Mesoscale	IgG titre	CSP-NANP, CSP-Cterm (Pf16), MSP-1, Pept-2, Pept-4, AMA-1, Pfs16, Pfs25	T0, T1
	Immunofluorescence assay	IFA titre	SPZ	T0, T1
B cells	Elispot	Number of Ag-spec. IgG secreting B cells/10^6^ B cells	AMA-1, CelTOS, CSP, MSP-1, SPZ	T0, T1
	Flow Cytometry	Frequency of activated B cells and subtypes (naïve, memory, memory switched, transitional, plasmablast)	SPZ	T0, T1, T2, T3
cTfh	Flow Cytometry	Frequency of activated cTfh and subtypes (cTfh1, cTfh2, cTfh17)	SPZ, CSP	T0, T1, T2, T3
CD4/CD8 T cells	Flow Cytometry	Frequency of activated CD4 T cells and subtypes (CXCR3^+^, CCR6^+^, Treg)	SPZ, CSP	T0, T1, T2, T3
		Frequency of activated CD8 T cells and subtypes (CXCR3^+^, CCR6^+^)	SPZ, CSP	T0, T1, T2, T3
Cytokines	Flow Cytometry ICS	Frequency of TNF-α and IFN-γ secreting CD4^+^ and CD8^+^ T cells	SPZ, CSP	T0, T1, T2, T3
	Mesoscale	pg/ml of IFN-γ, TNF-α, IL-1β, IL-10, IL-2, IL-4, IL-6, IL-8, IL-12p70, IL-13 produced by 10^6^ cells	AMA-1, CelTOS, CSP, MSP-1, hSPZ, lSPZ, TRAP	T0, T1

**Table 2 vaccines-10-00124-t002:** Immune measures predictive of protection status.

Immune Measure ^a^	Stimulation	Time Point	Relative Weights
TNFα secreting CD4^+^ T cells	SPZ	T1	100
TNFα secreting CD4^+^ T cells	SPZ	T0	79.3
Transitional B cells CD19^+^CD38^hi^CD24^hi^CD69^+^	SPZ	T1	60.6
TNFα secreting CD8^+^CXCR3^−^CCR6^−^ T cells	CSP	T0	55.9
TNFα secreting CD8^+^CXCR3^−^CCR6^−^ T cells	CSP	T1	47.2
IFNγ secreting CD8^+^CCR6^+^ T cells	CSP	T1	18.3
TNFα and IFNγ secreting CD4^+^CXCR3^−^CCR6^−^ T cells	CSP	T1	11.2

^a^ Immune measures and their relative variable importance to predict protection with an accuracy of 88% (κ = 0.64).

## Data Availability

All data for this study are contained within the manuscript and the supplementary materials. The scripts for running the analyses are deposited and are publicly available at https://github.com/BHSAI/IMRAS (accessed on 12 January 2022).

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
