# Peer review of "Immunoprofiling Identifies Functional B and T Cell Subsets Induced by an Attenuated Whole Parasite Malaria Vaccine as Correlates of Sterile Immunity"

_vaccines, 2022, doi:10.3390/vaccines10010124_

Round 1

Reviewer 1 Report

This manuscript, "Immunoprofiling Identifies Functional B and T Cell Subsets Induced by an Attenuated Whole Parasite Malaria Vaccine as Correlates of Sterile Protection," by Mura et al is, in the opinion of this reviewer, of high quality both in execution and presentation. It is acceptable for publication as is.

One small thing to look at would be to look over the appreciations to make sure that every single one is defied (e.g., CelTos), so that the non-specialist can more easily navigate the material.

Author Response

We diligently revised the manuscript and added definitions for abbreviations and edited to eliminate jargon. Reviewer 3 requested a formal section for abbreviations which we provided.

Reviewer 2 Report

In the present study, Mura et al used a combined immunoprofiling and machine learning approach to establish the immunological landscape of malaria-specific adaptive immune responses. The immunization was achieved via mosquito bites with radiation-attenuated Plasmodium falciparum sporozoite. The author identified several types of previously unidentified TNFα- and/or IFNγ-secreting T cells as well as functional B cells as indication of protection, showcasing the effectiveness of the approach used in this study, especially that of the recently developed computational analysis method.

Overall I think this work is well-performed and presented, and is suitable for publication in Vaccines. The statistics analyses in the manuscript are expertly performed. The finding is interesting in that previously unidentified cellular response to the immunization could be identified from combined immunoprofiling and machine learning approach, which highlight the application of modern computational tools in data analysis.

Author Response

Thank you for your positive feedback and the appreciation for our work. 

Reviewer 3 Report

The authors set out to identify through immunoprofiling the markers of protective immunity to malaria. To achieve this objective they used IMRAS ie immunization of human volunteers via mosquito bites with radiation attenuated sporozoites of Plasmodium falciparum (RAS)  Pf SPZ. A cohort of 21 volunteers were immunised following the IMRAS protocol and 5 controll subjects were mock-immunised,. All participants in the study were naive to malaria ie had never been exposed to malaria. Following the immunization a number of immunological parameters were measures and compared at different time points. The authors concluded that that the use of machine learning approaches allowed them to identify new parameters of immune protection. This is an important finding since comparison of immune parameters can be subjective depending on what cut-off point is chosen to delineate between a positive and negative response.

This  not withstanding there are a number of issues which need to be addressed before the paper can be accepted for publication and I mention them chronologically as they appear on the manuscript.

  1. The title talks of 'sterile protection' instead of 'sterile immunity', which is the consecrated term. The authors should either use the term sterile immunity or define the sterile protection in a footnote.
  2. the abstract should clearly name those new parameters of protective immunity which machine leaning enable them to identify (cf lineS 532-536)
  3. Many abbreviations are used in the text as if they were universally accepted. the authors should list the abbreviations and their meanings in a footnote to facilitate reading.
  4. The authors assume that they immunization protocol (IMRAS) induces sterile protection but have not provided the evidence to support this.They simply make reference to a previously published paper (ref 25). It is desirable that this evidence is summarized either in the Introduction or Discussion. Specifically, did IMRAS protect the immunized from developing parasitimia and malaria symptoms?
  5. The main focus of this paper is the ability of machine learning to delineate new T and B cell subsets involved in protective immunity, yet the description of this method is cryptic.The authors need to give more details of the method to enable the duplication of the results. In particular they should give references for the software employed.
  6. The sample size is small( 21 vaccine group vs 5 controls) Therefore, the authors should be less categorical in their conclusions.
  7. The role of neutralizing antibodies in immune protection to malaria is well known.The authors should discuss the inability of their method to detect the role of antibodies in IMRAS induced immunity to malaria.In conclusion the paper can be accepted after the authors have satisfactorily addressed the issues raised above.
